# Reduced-Order Model Based on Volterra Series for Aerodynamics of the Bridge Deck Section and Flutter Critical Wind Speed Prediction

**Ziran Wei, Zhiwen Liu \* and Fawei He**

Hunan Provincial Key Lab for Wind & Bridge Engineering, College of Civil Engineering, Hunan University, Changsha 410082, China
\* Correspondence: zhiwenliu@hnu.edu.cn; Tel.: +86-1397-588-0715

**Featured Application: This paper provides a valuable theoretical basis for the post-flutter response calculation that considers the vertical–torsional coupling. The aeroelastic analysis method proposed in this paper, based on the Volterra ROM and Newmark-*β* method, simplifies the process of flutter critical wind speed predictions; it requires engineers to only identify four Volterra first-order kernels. This method could be used to effectively solve flutter problems in long-span bridges.**

**Abstract:** This study proposes a novel reduced-order model (ROM), based on the Volterra series, for the aerodynamic force of the bridge deck section. Moreover, the ROM of the aerodynamic force of the streamlined box girder section of the Great Belt East Bridge (GBEB) is identified with computational fluid dynamic (CFD) simulations. Furthermore, an analysis method combining ROM aerodynamic force and Newmark-*β* integration is established to simulate the aeroelastic responses of the bridge deck section. Finally, the wind-induced vibration responses of the GBEB section are calculated near the flutter critical wind speed based on the Volterra series-based aeroelastic analysis and the fluid–structure interaction (FSI) numerical simulations in ANSYS Fluent, respectively. Moreover, to verify the applicability of the proposed method, the aeroelastic responses of the main deck section with the crash barriers of Lingdingyang Bridge (LDYB) are also simulated via the Volterra model and Newmark-*β* integration near the flutter critical wind speed. The results show that the first-order truncated Volterra model established in this study can accurately capture the aerodynamic response of the main girder, and the results are in good agreement with those of the CFD numerical simulation under forced vibration. Furthermore, the proposed method combined with ROM aerodynamic force and Newmark-*β* integration can effectively calculate the FSI of the bridge girder. The numerical results of the flutter critical wind speed and flutter frequency of GBEB and LDYB are consistent with the numerical results by the FSI method based on ANSYS Fluent and the existing numerical and experimental results, respectively.

**Keywords:** reduced-order model (ROM); Volterra series; streamlined box girder section; wind-induced vibration; flutter critical wind speed

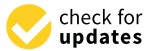



## 1. Introduction

As one of the wind-induced vibration problems, flutter may cause large-amplitude divergent vibrations of the main deck of bridges, resulting in structural damage and serious consequences. Four months after the completion of the original Tacoma Narrows Bridge, a sizeable torsional vibration occurred for nearly 70 min under a wind speed of 18 m/s, and the bridge eventually collapsed (Farquharson [1]). However, the existing bridge aeroelastic analysis widely adopts the linear flutter theory provided by Scanlan (Scanlan et al. [2]). The linear flutter theory is based on the assumption of linearity and small disturbances, which ignores the large-amplitude and large angle-of-attack (AOA)

motions of long-span bridges and the significantly blunted aerodynamic shape of the deck (Patil et al. [3]). The above situation may cause significant flow separation and reattachment around the bridge girder section, leading to apparent nonlinearity of aerodynamics and thus, the inaccuracy of the linear flutter theory (Scanlan [4]). Due to the nonlinear effects of the aerodynamic force on large-amplitude vibrations or large AOAs, many researchers have proposed different aerodynamic reduced-order models (ROMs) to consider the nonlinear effect of aerodynamic force (Dowell [5]; Silva [6]; Silva et al. [7]), such as the harmonic balanced (HB) method, proper orthogonal decomposition (POD) method and Volterra series. Hall et al. [8] utilized the HB method for the first time to model unsteady nonlinear flows in turbomachinery, while Romanowski [9] utilized the POD method to construct an unsteady aerodynamic and aeroelastic model for two-dimensional isolated airfoils in a compressible flow. However, these methods cannot fully consider the fluid memory effects in the fluid–structure interaction (FSI). The fluid memory indicates that both the historical structural motion and the flow field fluctuations have an impact on the current flow field state (Wu et al. [10]). Therefore, the Volterra series is more suitable for solving the complex interaction between the structure and the flow field in the aeroelastic problem, such as vortex-induced vibrations and flutter.

Wiener [11] utilized the Volterra series theory to analyze a nonlinear system and proposed its discrete form. The identification of the kernels is a crucial step in constructing the Volterra model. Subsequently, Clancy et al. [12] and Tromp et al. [13] proposed the impulse response method and the step response method to identify the Volterra kernels, respectively. Later, Silva [14] proposed a multi-input–multi-output (MIMO) identification method, which significantly improved the efficiency of the Volterra kernel identification. The Volterra series-based aerodynamic ROM has been widely utilized to solve nonlinear aerodynamic problems in aeronautical engineering and bridge engineering. Wu et al. [15] utilized a sparse third-order Volterra model to simulate the nonlinear aerodynamics of the bridge girder section, and the robustness of the sparse Volterra model was verified. Liu et al. [16] predicted the unsteady aerodynamics of a flapping wing by a ROM based on the Volterra series. Henrik et al. [17] utilized a Laguerrian expansion basis to simplify the least-squares problem by parameterizing the kernels. Ali et al. [18] presented a nonlinear time-domain analysis framework based on Volterra series to simulate the buffeting response of long-span bridges. Ruiz et al. [19] identified a finite memory Volterra model to predict the characteristics of forces and moments produced by the flapping wing. Xu et al. [20] simulated the response of vortex-induced vibration of the bridge deck by a Volterra series-based ROM. Li et al. [21] proposed a Volterra model to predict the aerodynamic forces of the blades induced by upstream wakes. Lin et al. [22] utilized the Fourier-transformed Volterra kernels to model the aeroelastic systems for flutter. Although there is some research focusing on nonlinear aerodynamics by utilizing the Volterra model, few studies on the prediction of the flutter response based on the Volterra series-based aerodynamic ROM have been found recently. Additionally, current aerodynamic self-excited force models are mainly used for single-degree-of-freedom flutter analysis. There is a lack of models considering the vertical–torsional coupling. Therefore, further investigation into the aeroelastic analysis of the bridge deck section is necessary. Moreover, taking into account the advantage of multiple-degrees-of-freedom and the convenient identification of the first-order kernel, it is of great value and significance for the development of long-span bridges to carry out the rapid analysis method research of two-degree-of-freedom flutter based on Volterra model.

In this study, a new aeroelastic analysis method is proposed by combining an aerodynamic Volterra series-based ROM with the Newmark-$\beta$ method, which takes into account the vertical–torsional coupling. Moreover, taking the main girder section of the Great Belt East Bridge (GBEB) as an example, the nonlinear aerodynamic ROM is compared with CFD numerical simulation to evaluate the aerodynamic response under forced vibration. Furthermore, the new aeroelastic analysis method is utilized to analyze the wind-induced vibration displacement responses of the stiffening girder section of the GBEB and the

Lingdingyang Bridge (LDYB) under different wind speeds to identify the flutter critical wind speed.

## 2. Volterra Model and FSI Method

### 2.1. Volterra Series Theory

The Volterra series theory was first proposed by Volterra in 1880 (Volterra [23]). According to the Volterra series theory, for any continuous-time nonlinear and time-invariant system, the relationship between the system input $u(t)$ and the system output $y(t)$ can be expressed as follows,

$$
\begin{aligned}
y(t) = h_0 &+ \int_0^t h_1(\tau_1)u(t - \tau_1)d\tau_1 + \int_0^t \int_0^t h_2(\tau_1, \tau_2)u(t - \tau_1)u(t - \tau_2)d\tau_1 d\tau_2 \\
&+ \cdots + \int_0^t \cdots \int_0^t h_n(\tau_1, \cdots, \tau_n)u(t - \tau_1) \cdots u(t - \tau_n)d\tau_1 \cdots d\tau_n
\end{aligned}
\tag{1}
$$

where $u(t - \tau_k)$ denotes the system input; $y(t)$ denotes the system output; $h_0$ represents the steady-state term; $h_1$ represents the first-order kernel; $h_n$ represents the higher-order term which describes nonlinear features of the system.

### 2.2. Volterra Kernel Identification Based on Impulse Function

The nonlinear aerodynamic force of the bridge girder section is approximately a time-invariant system. It has been proven that the establishment of a truncated low-order nonlinear model, namely a first-order truncated Volterra series, can adequately describe the nonlinear effects (Wu et al. [24]; Wu et al. [25]). For a first-order truncated Volterra series, Equation (1) can be rewritten as Equation (2), and the first-order kernel can be directly expressed as the output response under the action of a unit impulse equation, that is, $h_1(t) = y_1(t)$. Such a first-order kernel is entirely linear. Therefore, Equation (3) is utilized to identify the first-order Volterra kernel. It can be seen that the first-order Volterra kernel obtained by Equation (3) considers the nonlinear effect of different amplitudes, which is different from the general linear kernel.

$$
y(t) = h_0 + \int_0^t h_1(\tau_1)u(t - \tau_1)d\tau_1
\tag{2}
$$

$$
h_1(t) = \frac{1}{\beta^2 - \beta}\{\beta^2[y_1(t) - h_0] - [y_\beta(t) - h_0]\}
\tag{3}
$$

where $y_1(t)$ indicates the system output under the unit-impulse response; $y_\beta(t)$ indicates the system output under $\beta$ times the unit-impulse response.

The smoothed-ramped function is usually utilized as the impulse equation for the Volterra kernel identification. There are three commonly used smoothed-ramped functions, namely the sinusoidal function, the polynomial function, and the exponential function. The expressions of the three smoothed-ramped functions are shown in Equations (4a)–(4c)

$$
V_{sin} = \begin{cases} \frac{\pi}{2}\frac{A}{T}\sin(\frac{\pi}{T}t) & 0 \le t < T \\ 0 & t \ge T \end{cases}
\tag{4a}
$$

$$
V_{poly} = \begin{cases} \frac{A}{T^3}\left(30t^2 - \frac{60}{T}t^3 + \frac{30}{T^2}t^4\right) & 0 \le t < T \\ 0 & t \ge T \end{cases}
\tag{4b}
$$

$$
V_{exp} = \begin{cases} A\exp\left\{-\left[b\left(t - \frac{T}{2}\right)\right]^2\right\} & 0 \le t < T \\ 0 & t \ge T \end{cases}
\tag{4c}
$$

where $A$ and $T$ denote the amplitude and duration of the impulse function, respectively.

It should be noted that there is a discontinuity in acceleration in the case of a sinusoidal function, and the displacement, velocity, and acceleration of the smoothed-ramped function based on the exponential function can only be approximately close to 0.0 at the beginning and end time. Therefore, the smoothed-ramped function based on the polynomial function is applied to identify the kernels in this study. Taking the heaving motion as an example, Figure 1 shows the structural displacement, velocity, and acceleration under a vertical pulse designed by Equation (4b).

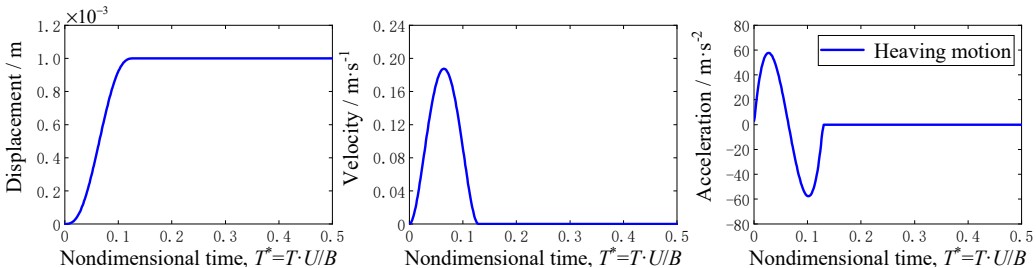

**Figure 1.** Characteristics of polynomial-basis smoothed-ramped impulse function.

### 2.3. Computational Approach

To verify the accuracy of the method in this study, the main deck section of GBEB is taken as an engineering background. The identification of the aerodynamic force of the main deck section without crash barriers is carried out by applying the aerodynamic force model proposed in this study. Moreover, the wind-induced vibration responses of the main deck for different wind speeds are analyzed with the FSI method proposed in this study.

Figure 2 shows the main deck section of the GBEB. As shown in Figure 2, the width and height of the main deck section are 31 m and 4.4 m, respectively. Furthermore, the shear center is 2.354 m from the bottom of the main deck section.

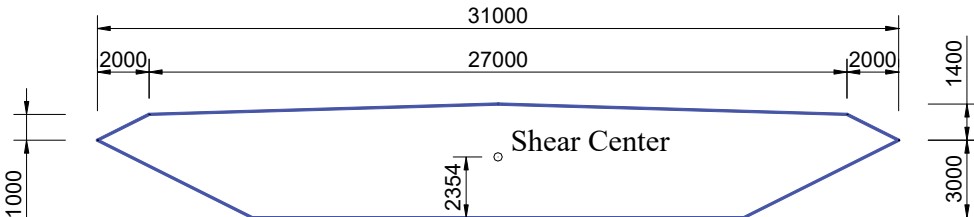

**Figure 2.** Girder cross-section of GBEB main span (unit: mm).

For incompressible viscous fluids, the mass conservation equation and the momentum conservation equation can be expressed as follows,

$$\nabla \cdot \boldsymbol{V} = 0 \tag{5}$$

$$\frac{\partial \boldsymbol{V}}{\partial t} + (\boldsymbol{V} \cdot \nabla)\boldsymbol{V} = \boldsymbol{f} - \frac{1}{\rho}\nabla p + \frac{\mu}{\rho}\nabla^2 \boldsymbol{V} \tag{6}$$

where $\rho$ denotes the fluid density; $\boldsymbol{V}$ denotes the velocity vector; $p$ denotes the pressure; $\boldsymbol{f}$ denotes the external force on the unit volume fluid, if gravity is considered, $\boldsymbol{f} = \rho g$; $\mu$ denotes the dynamic viscosity.

Considering the blocking ratio of the main deck section, it is determined empirically that the calculation domain of the main deck section adopts a rectangular area of $35\,B \times 20\,B$ ($B$, the width of the main girder section), as shown in Figure 3. The boundary conditions are defined as follows: the left side of the computational domain is set as velocity inlet, the right side of the computational domain is set as pressure outlet, the main deck section is set as a non-slip wall, and the upper and lower sides of the computational domain are set as symmetry boundaries. The block meshing method was adopted to discrete the

computational domain, namely the rigid domain, the dynamic grid domain and the static grid domain, respectively.

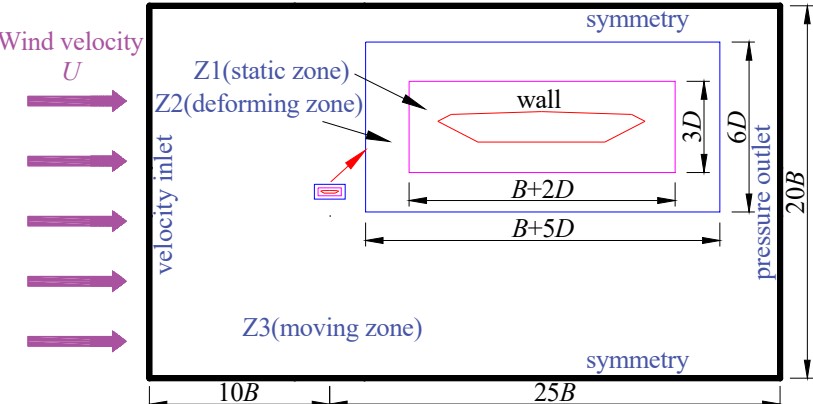

**Figure 3.** Computational domain, boundary conditions, and blocks.

The geometric scale ratio of the main deck section is determined as 1:80. The height of the first layer of the grid is 1.0 cm, which is about $3 \times 10^{-4}$ $B$, and the total grid number is 112,706. Figure 4 indicates the local division of the grids near the main deck section. The shear stress transport (SST) $k$-$\omega$ turbulence model is adopted in the Navier–Stokes equations in the time domain (Menter [26]). Considering that the Volterra kernel identification needs to ensure a certain continuity of the smoothed-ramped function, the time step of numerical simulation needs to be small enough. Therefore, the time step is defined as $\Delta t$ = 0.0005 s in this study. The wind speed ratio is 1:8, the incoming wind speed is $U$ = 5 m/s, and the corresponding wind speed at the bridge is 40 m/s, which is close to the flutter critical wind speed of the GBEB for subsequent flutter analysis.

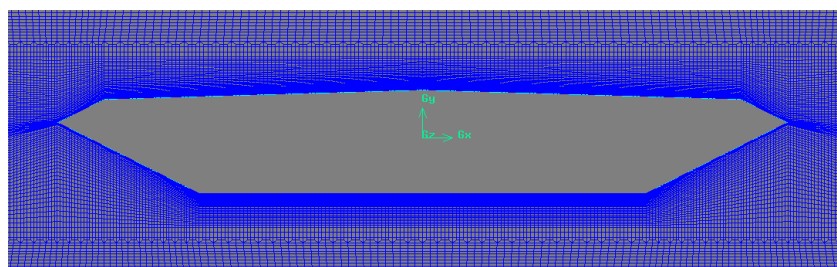

**Figure 4.** Mesh near the GBEB girder section model.

*2.4. Aeroelastic Analysis Based on the Volterra Model and Newmark-β Method*

The vibration differential equations of a two-degree-of-freedom system shown in Figure 5 can be expressed as

$$\ddot{y}(t) + 2\xi_h \omega_h \dot{y}(t) + \omega_h^2 y(t) = F_L(t)/m \tag{7a}$$

$$\ddot{\alpha}(t) + 2\xi_\alpha \omega_\alpha \dot{\alpha}(t) + \omega_\alpha^2 \alpha(t) = M_\alpha(t)/I_m \tag{7b}$$

where $m$ and $I_m$ represent the mass and torsion mass moment of inertia of the structure, respectively; $\zeta_h$ and $\zeta_\alpha$ represent the vertical bending and torsional damping radio, respectively. $\omega_h$ and $\omega_\alpha$ represent the circular frequency of vertical and torsional vibration, respectively. $F$ and $M$ represent the vertical and torsional load, respectively; $y$ and $\alpha$ represent the vertical displacement and the torsional displacement, respectively.

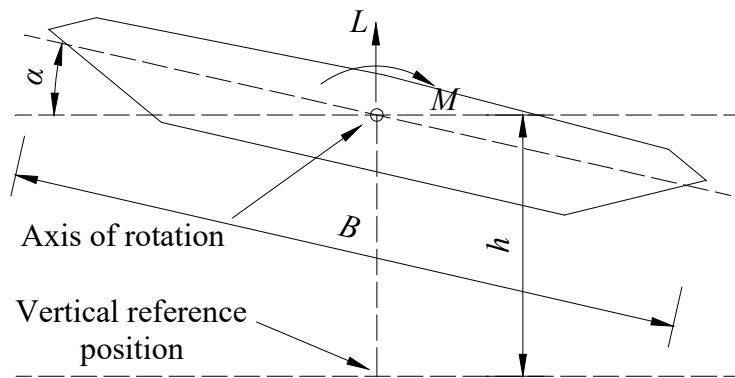

**Figure 5.** Aerodynamic forces and displacements of the girder section model.

The Newmark-$\beta$ method assumes that the structural displacement, velocity, and acceleration of the discrete-time system can be expressed by introducing coefficients $\beta$ and $\gamma$ as follows

$$\overline{k}y_{t+\Delta t} = \overline{F}_{t+\Delta t} \tag{8a}$$

$$\dot{y}_{t+\Delta t} = \frac{\gamma}{\beta \Delta t}(y_{t+\Delta t} - y_t) + (1 - \frac{\gamma}{\beta})\dot{y}_t - (\frac{\gamma}{2\beta} - 1)\ddot{y}_t \Delta t \tag{8b}$$

$$\ddot{y}_{t+\Delta t} = \frac{1}{\beta \Delta t^2}(y_{t+\Delta t} - y_t) - \frac{1}{\beta \Delta t}\dot{y}_t - (\frac{1}{2\beta} - 1)\ddot{y}_t \tag{9}$$

where the expressions of $\overline{k}$ and $\overline{F}_{t+\Delta t}$ are shown in Equation (9)

$$\overline{k} = k + \frac{m}{\beta \Delta t^2} + \frac{\gamma c}{\beta \Delta t} \tag{9a}$$

$$\overline{F}_{t+\Delta t} = F_{t+\Delta t} + m\left\{\frac{y_t}{\beta \Delta t^2} + \frac{\dot{y}_t}{\beta \Delta t} + (\frac{1}{2\beta} - 1)\ddot{y}_t\right\} + c\left\{\frac{\gamma y_t}{\beta \Delta t} + (\frac{\gamma}{\beta} - 1)\dot{y}_t + (\frac{\gamma}{2\beta} - 1)\Delta t\ddot{y}_t\right\} \tag{9b}$$

In the aerodynamic model based on the Volterra series, the aerodynamic lift force coefficient and torsion moment coefficient are related to the velocity of heaving and pitch motion. Therefore, the aerodynamic self-excited force can be expressed as follows

$$L(t) = \frac{1}{2}\rho U^2 B C_L(t) \tag{10a}$$

$$M(t) = \frac{1}{2}\rho U^2 B^2 C_M(t) \tag{10b}$$

$$C_L(t) = \int_0^t h_{1uL}(\tau_1)\dot{u}(t - \tau_1)dt + \int_0^t h_{1\theta L}(\tau_1)\dot{\theta}(t - \tau_1)dt + C_{L0} \tag{11a}$$

$$C_M(t) = \int_0^t h_{1uM}(\tau_1)\dot{u}(t - \tau_1)dt + \int_0^t h_{1\theta M}(\tau_1)\dot{\theta}(t - \tau_1) + C_{M0} \tag{12}$$

In this case, the Volterra series-based ROMs are models of four first-order Volterra kernels to be identified. Accordingly, due to the advantage of fewer parameters to be solved, the computational cost to establish the ROMs can be greatly reduced compared with flutter derivative theory and some other flutter analysis methods, e.g., aerodynamic describing-function-based models (Zhang et al. [27]). Moreover, the vertical–torsional-coupling effect has been considered, significantly improved over single-degree-of-freedom (SDOF) torsional flutter simplified models (e.g., Gao et al. [28]).

Figure 6 shows the working process of the aeroelastic numerical simulation for flutter analysis established in this study based on the proposed aerodynamic model via the Volterra

series and Newmark-$\beta$ method. Moreover, the aerodynamic force at time $t$ is calculated from the structural displacement at time $t - \Delta t$, and then, the structural displacement at time $t$ is calculated from the aerodynamic force at time $t$ by the Newmark-$\beta$ method.

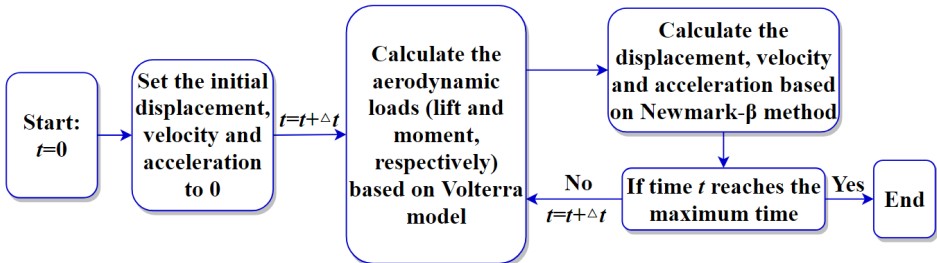

**Figure 6.** Schematic representation of the numerical simulation based on the Volterra model.

## 3. Validations and Applications

### 3.1. Validations

The vertical unit displacement is defined as 0.001 m (0.0025 $B$), and the torsional unit displacement is defined as $0.5°$. Using the CFD numerical simulation method, the aerodynamic responses of the scaled-girder section of the GBEB, under the unit pulse excitation and twice unit pulse excitation, are calculated under the wind attack angle of $0°$ and the calculation wind speed of 5 m/s, respectively. Moreover, Equation (3) is utilized to calculate the first-order kernel. The duration of the impulse function is defined as $T = 0.01$ s, corresponding to the nondimensional time of $T^* = T \cdot U/B \approx 0.13 < 0.2$ (Wu et al. [25]). Figure 7 shows the first-order kernel of the lift-force coefficient $C_L$ and torsion-moment coefficient $C_M$ of the main deck section of GBEB in the heaving and pitch motion. While the first-order kernel responds significantly to pulsed excitation, it quickly approaches zero within a few non-dimensional time steps after the excitation ends.

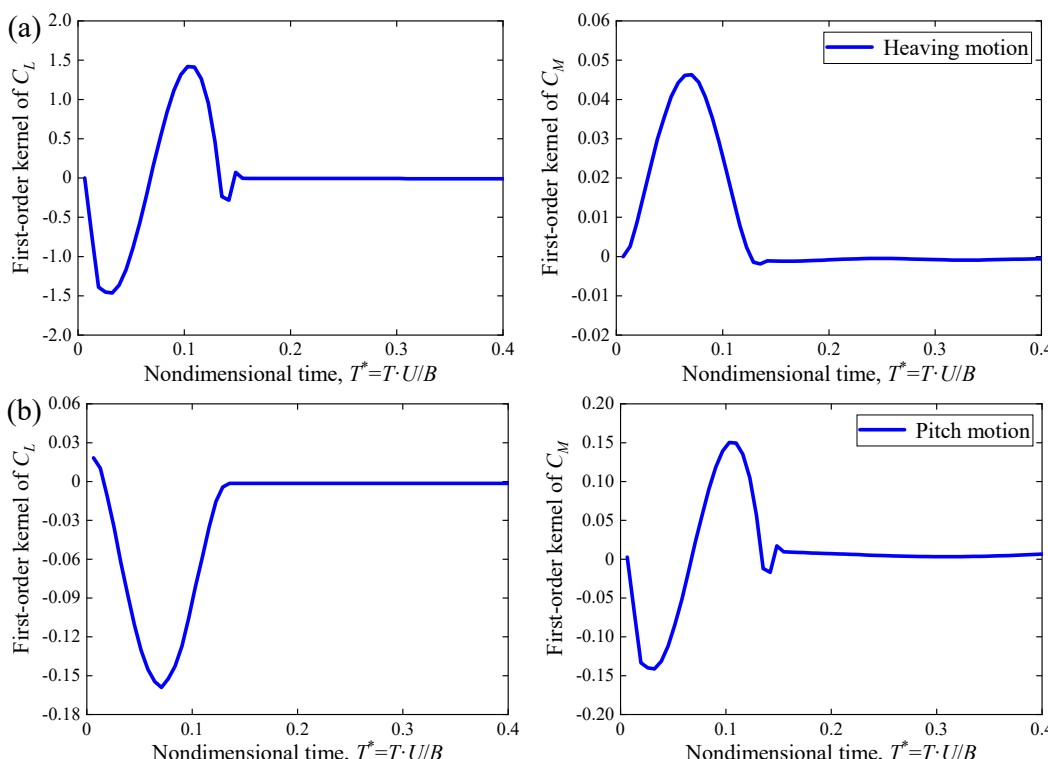

**Figure 7.** The first-order kernels of lift and moment coefficients (**a**) in the heaving motion; (**b**) in the pitch motion.

To verify the accuracy of the established aerodynamic model of the main deck section of GBEB based on the Volterra series, the Volterra model is utilized to calculate the aerodynamic force coefficients under multiple sinusoidal displacements. The expression of the heaving and pitch motion is shown in Equation (12). The aerodynamic ROM simulation calculation is carried out at 0° wind attack angle. The results are compared with those of CFD numerical simulation under six operating conditions with vertical displacements of about 0.01 *B* and 0.10 *B*, torsional angles of 3° and 6°, and frequencies of 1.0 Hz and 4.0 Hz, respectively. Tables 1 and 2 show the parameters of each operating condition and the aerodynamic calculation results of the two methods.

$$y = y_h \sin(2\pi f_h t) \tag{12a}$$

$$\theta = \theta_\alpha \sin(2\pi f_\alpha t) \tag{12b}$$

**Table 1.** Parameters of heaving motion and comparisons of results.

| Case | $y_h$/m | $f_h$/Hz | Root Variance of Force Coefficient | First-Order Approximation | Numerical Simulation | Relative Error, $\delta$/% |
|------|---------|----------|-------------------------------------|---------------------------|----------------------|----------------------------|
| 1 | 0.004 | 1 | $C_L{}'$ | 0.0145 | 0.0140 | 3.57 |
|   |       |   | $C_M{}'$ | 0.0038 | 0.0036 | 5.55 |
| 2 | 0.04  | 1 | $C_L{}'$ | 0.1447 | 0.1396 | 3.65 |
|   |       |   | $C_M{}'$ | 0.0378 | 0.0359 | 5.29 |
| 3 | 0.04  | 4 | $C_L{}'$ | 0.4990 | 0.4796 | 4.05 |
|   |       |   | $C_M{}'$ | 0.1209 | 0.1252 | 1.91 |

Note: $y_h$ = oscillation amplitude; $f_h$ = oscillation frequency; relative error $\delta = |Y_1 - Y_2| / |Y_1| \times 100\%$; $Y_1$ = the results of first-order approximation; $Y_2$ = the results of numerical simulation.

**Table 2.** Parameters of pitch motion and comparisons of results.

| Case | $\theta_\alpha$/° | $f_\alpha$/Hz | Root Variance of Force Coefficient | First-Order Approximation | Numerical Simulation | Relative Error, $\delta$/% |
|------|-------------------|---------------|-------------------------------------|---------------------------|----------------------|----------------------------|
| 4 | 3 | 1 | $C_L{}'$ | 0.1460 | 0.1474 | 0.95 |
|   |   |   | $C_M{}'$ | 0.0371 | 0.0392 | 5.36 |
| 5 | 6 | 1 | $C_L{}'$ | 0.2919 | 0.2963 | 1.48 |
|   |   |   | $C_M{}'$ | 0.0743 | 0.0786 | 5.47 |
| 6 | 6 | 4 | $C_L{}'$ | 0.2761 | 0.2910 | 5.12 |
|   |   |   | $C_M{}'$ | 0.0919 | 0.0871 | 5.51 |

Note: $\theta_\alpha$ = oscillation amplitude; $f_\alpha$ = oscillation frequency; relative error $\delta = |Y_1 - Y_2| / |Y_1| \times 100\%$; $Y_1$ = the results of first-order approximation; $Y_2$ = the results of numerical simulation.

Figure 8 indicates the time histories of the aerodynamic force coefficients calculated by the two methods. It can be seen from Figure 8 that the aerodynamic system of the girder section of GBEB has good identification results from the Volterra first-order model in both the heaving and pitch motions, and the overall error is almost within 10%. When the forced vibration amplitude (the vertical amplitude is less than 0.1 *B* and the torsional amplitude is less than 6°) or frequency (less than 4 Hz) is changed in a limited range, the error of the simulation of the Volterra first-order model does not increase significantly compared with the CFD numerical simulation results. Therefore, it can be considered that the Volterra first-order model established in this section can effectively simulate the aerodynamic force of the girder section of GBEB.

### 3.2. Numerical Results of Wind-Induced Vibration Responses

To further verify the accuracy of the ROM established in this study, the wind-induced vibration responses of the main deck section of the GBEB for different wind speeds near the flutter critical wind speed are calculated by ROM and FSI, respectively. The main parameters of the main deck section of GBEB are given in Table 3.

In the FSI numerical simulation, the turbulence intensity is set to 0.5%, the turbulent viscosity ratio is set to 2, and the initial uniform flow field turbulence characteristic is set for the whole calculation domain. The incoming wind speed is set to 4.735 m/s, 5 m/s, and 5.625 m/s, respectively, corresponding to the wind speeds of 35 m/s, 40 m/s, and 45 m/s for the prototype main deck section, respectively.

Figure 9 shows the numerical results of wind-induced vibration responses of the main deck section by FIS numerical simulation for different incoming wind speeds, respectively. As shown in Figure 9, when the incoming wind speed is 35 m/s for the prototype main deck section, the torsional displacement and vertical displacement of the prototype girder have no obvious flutter divergence phenomenon. However, when the incoming wind speed is 40 m/s for the prototype main deck section, the torsional displacement and vertical displacement of the main deck section are stable, and the displacement attenuation rate is 0.05%, which is close to 0, which can be considered close to the flutter critical wind speed. Furthermore, when the incoming wind speed is 45 m/s for the prototype main deck section, the torsional displacement and vertical displacement of the girder have an obvious flutter divergence phenomenon. Therefore, it can be concluded that the flutter critical wind speed of the main girder of GBEB is about 40 m/s.

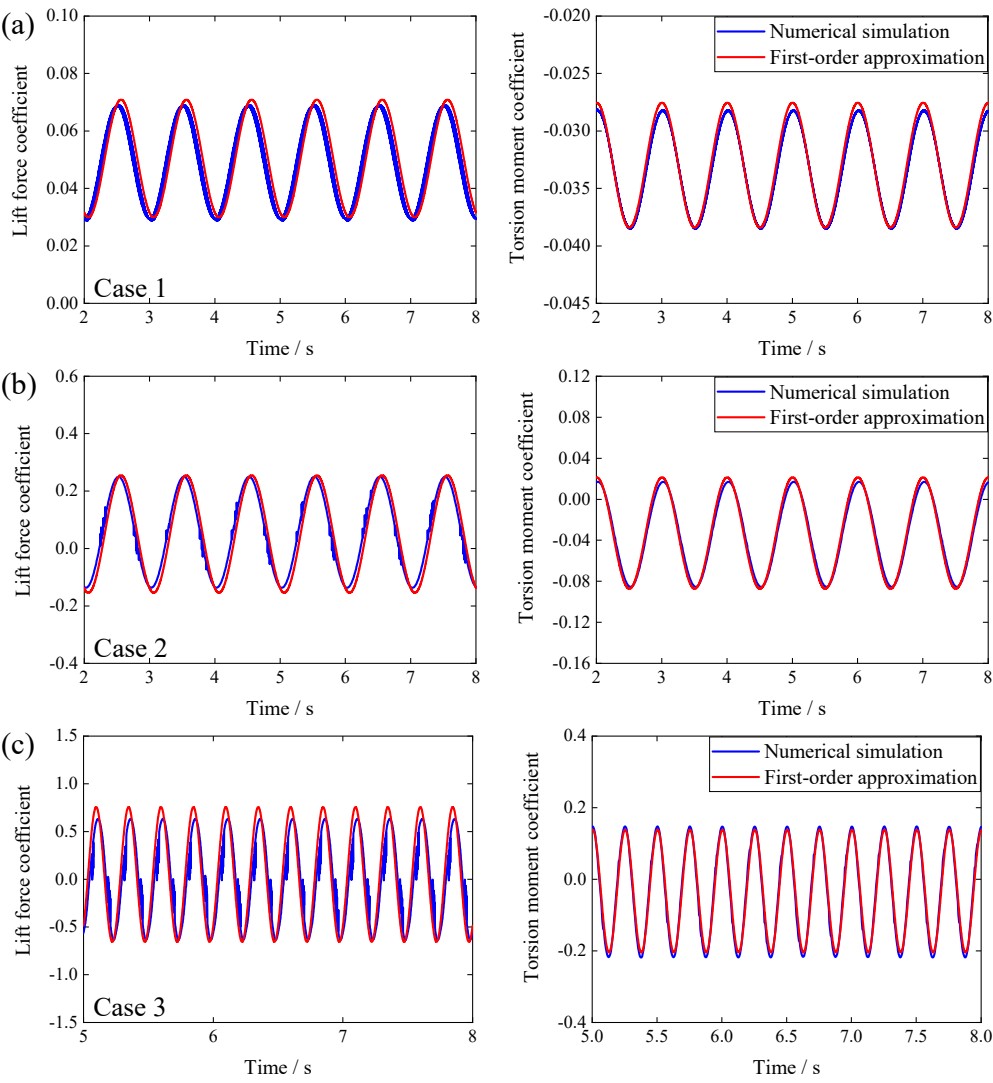

**Figure 8.** *Cont.*

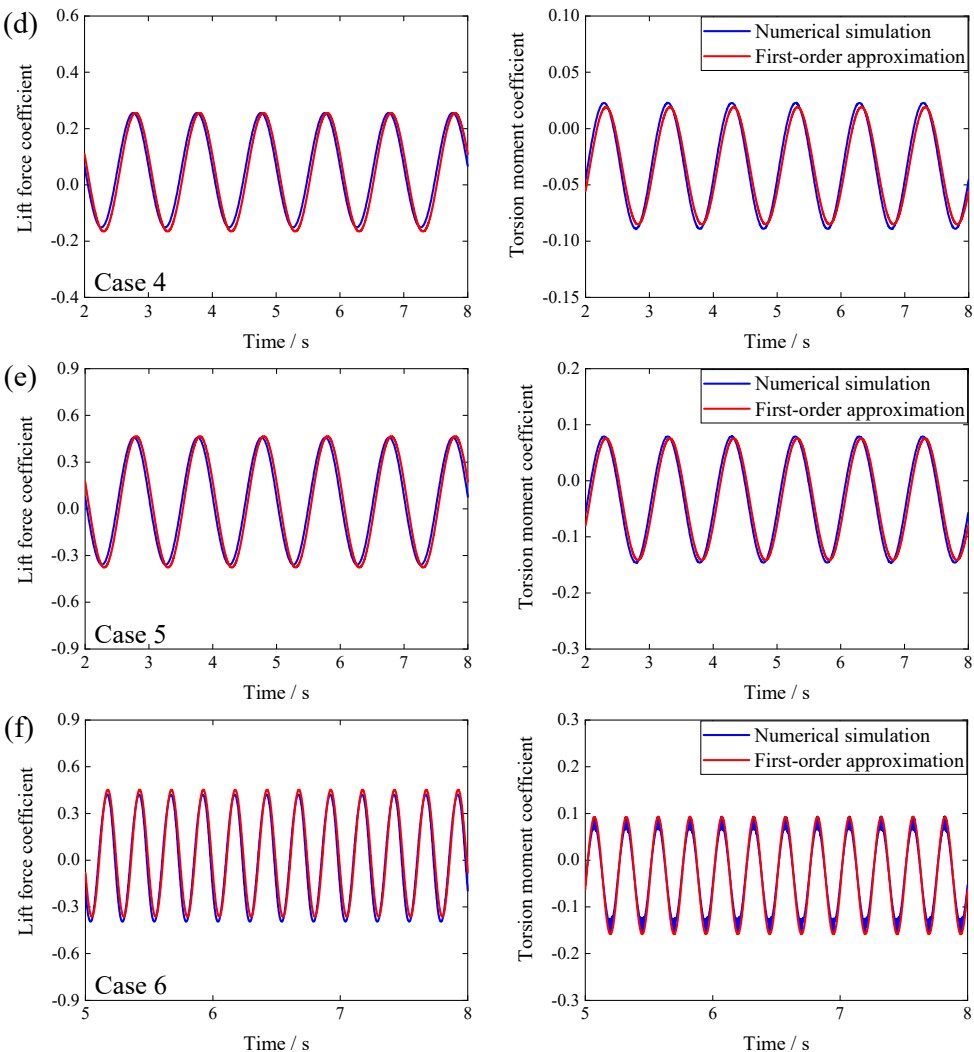

**Figure 8.** Comparisons of results of force coefficients (**a**) in Case 1; (**b**) in Case 2; (**c**) in Case 3; (**d**) in Case 4; (**e**) in Case 5; and (**f**) in Case 6.

**Table 3.** Main parameters of the main deck section of GBEB.

| Parameters | Unit | Prototype Main Deck Section | Scale Ratio | Main Deck Section MODEL |
|---|---|---|---|---|
| Width, $B$ | m | 31.0 | 1:80 | 0.3875 |
| Height, $H$ | m | 4.4 | 1:80 | 0.0550 |
| Mass per unit length, $m$ | kg/m | $17.8 \times 10^3$ | $1:80^2$ | 2.781 |
| Torsion mass moment of inertia per unit length, $I$ | kg·m$^2$/m | $2.173 \times 10^6$ | $1:80^4$ | 0.053 |
| Vertical bending frequency, $f_h$ | Hz | 0.099 | 10:1 | 0.99 |
| Torsional frequency, $f_\alpha$ | Hz | 0.186 | 10:1 | 1.86 |
| Vertical bending–damping ratio, $\zeta_h$ | % | 0.5 | / | 0.5 |
| Torsional damping ratio, $\zeta_\alpha$ | % | 0.5 | / | 0.5 |
| Wind velocity, $U$ | m/s | / | 1:8 | / |

The structural dynamic parameters of the main girder of the GBEB in Table 3 are applied to the FSI program, based on the Volterra model and Newmark-$\beta$ method. It should be noted that the first-order kernel function of the Volterra series of the aerodynamic force of the main deck section model is identified for the wind speed of 5 m/s. Here, assuming that the first-order kernel of the Volterra series for the wind speed of 5 m/s is

equal to that for the wind speeds of 4.375 m/s, and 5.625 m/s, respectively. Figure 10 shows the numerical results of wind-induced vibration responses of the main deck section by ROM method for incoming wind speeds of 35 m/s, 40 m/s, and 45 m/s for the prototype main deck section, respectively.

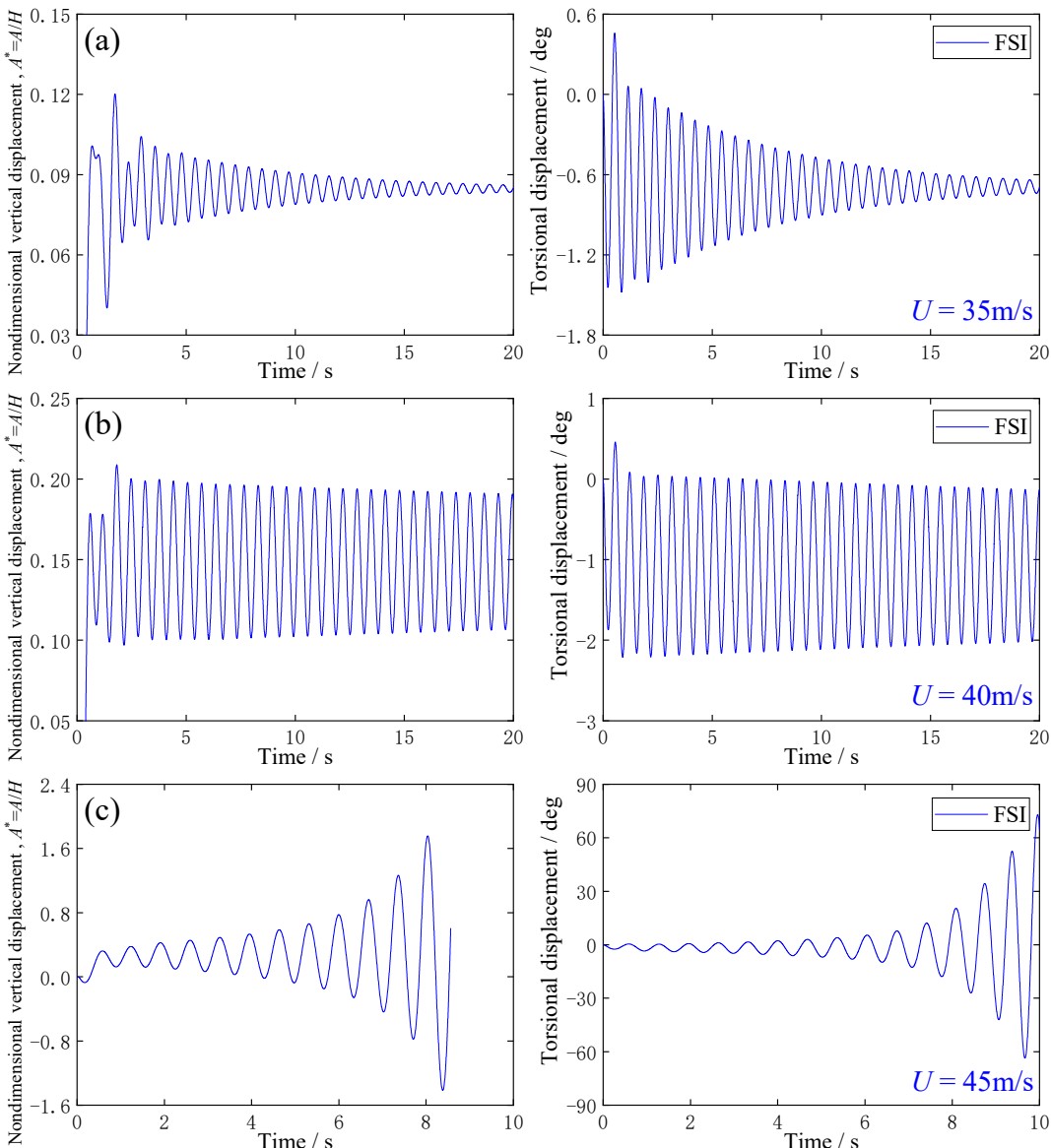

**Figure 9.** Vertical and torsional displacements of main deck section of GBEB, based on the FSI method for incoming wind speeds of (**a**) 35 m/s, (**b**) 40 m/s, and (**c**) 45 m/s for the prototype bridge, respectively.

As shown in Figure 10, when the incoming wind speed is 35 m/s for the prototype main deck section, the torsional displacement and vertical displacement of the prototype girder have no obvious flutter divergence phenomenon. However, when the incoming wind speed is 40 m/s for the prototype main deck section, the torsional displacement and vertical displacement of the girder show a trend of slow divergence. Furthermore, when the incoming wind speed is 45 m/s for the prototype main deck section, the torsional displacement and vertical displacement of the girder have an obvious flutter divergence phenomenon. The calculation result of the flutter critical wind speed of the main girder section of GBEB obtained by the ROM method is about 40 m/s. Meanwhile, Figures 9 and 10 have consistency for the flutter analysis results.

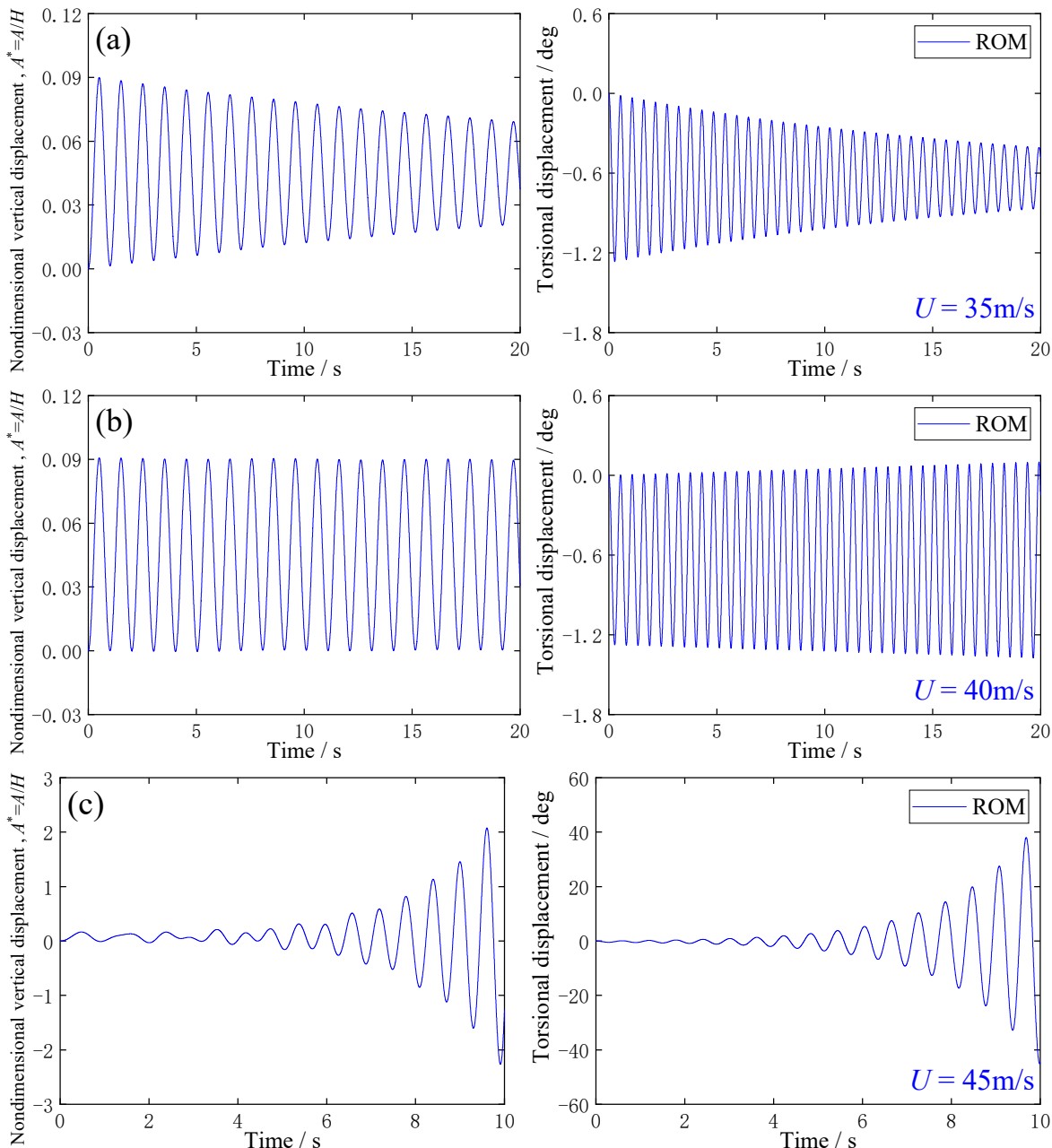

**Figure 10.** Vertical and torsional displacements of the main deck section of GBEB, based on the ROM method under incoming wind speeds of (**a**) 35 m/s, (**b**) 40 m/s, and (**c**) 45 m/s for the prototype bridge, respectively.

Table 4 summarizes the flutter critical wind speed and flutter frequency results of the main girder of the GBEB calculated by FSI numerical simulation based on CFD and the aeroelastic analysis method in this study and other examples of research. Table 4 represents that the flutter critical wind speed of the girder of GBEB obtained by different methods are similar, and the flutter frequency is also in good agreement. Among the different methods, the numerical simulation method has a long calculation period. Although the wind tunnel test method has a short test period, there is complicated preparation work in the early stage of the test, which requires more workforce and material resources. In contrast, the aeroelastic analysis method based on the Volterra series nonlinear aerodynamic model and the Newmark-$\beta$ method established in this study has the shortest calculation time and the least parameters to be identified. However, it should be noted that the wind tunnel test includes auxiliary structures such as railings, which reduces the streamlined degree of the

girder section. Therefore, the experimental result of the flutter critical wind speed is lower than the numerical results.

**Table 4.** Comparisons of flutter results of the girder of GBEB.

| Method | Flutter Critical Wind Speed (m/s) | Flutter Frequency (Hz) |
| --- | --- | --- |
| Finite volume method (Fujiwara [29]) | 40.5 | 0.160 |
| Discrete-vortex model (Walther [30]) | 37.6 | 0.165 |
| Wind tunnel test (Poulsen et al. [31]) | 36.0 | 0.163 |
| FSI (present) | 40.0 | 0.164 |
| ROM (present) | 35.0~40.0 | 0.160 |

It should be noted that the aeroelastic analysis method developed in this study uniformly utilizes the first-order kernel corresponding to the actual girder wind speed of 40 m/s in the calculation. The aeroelastic analysis is carried out under other wind speeds by assuming that the first-order kernels within a specific wind speed range are equal. Considering that there may be a particular deviation in the wind speed results, this paper only gives the wind speed range and does not calculate the specific flutter critical wind speed.

*3.3. Practical Application*

Since the main girder sections of most actual bridges are more significantly blunted than that of GBEB, this paper takes LDYB, a suspension bridge with a main span of 1666 m under construction in Guangdong, China, as another engineering background to further study the accuracy of using the Volterra series-based ROM model to describe the aerodynamic nonlinearity of the main deck section with crash barriers. LDYB is an essential part of the Shenzhen–Zhongshan Link Project, which is one of China's critical cross-river projects and is of great significance to their economic development. Figure 11 shows the original main deck section of LDYB. As shown in Figure 11, the width and height of the main deck section are 49.7 m and 4.0 m, respectively. Compared with the case of the GBEB, the main deck section of LDYB includes sidewalk outriggers, crash barriers, and inspection rails, where the nonlinear aerodynamic will be more foreseeably significant.

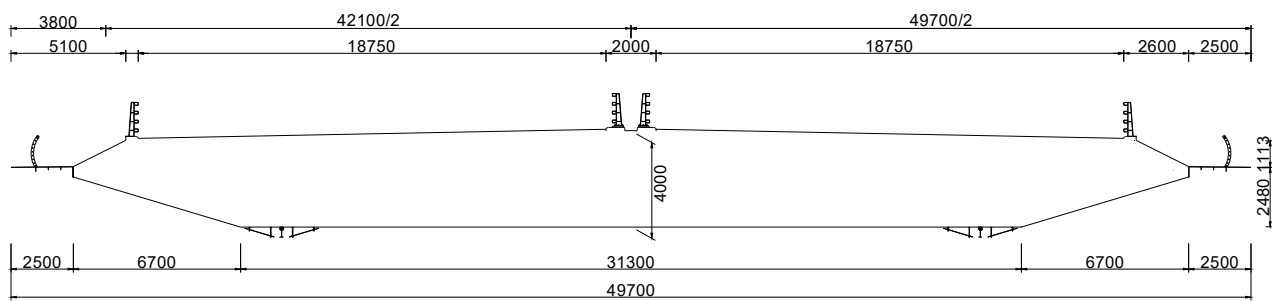

**Figure 11.** Girder cross-section of LDYB main span (unit: mm).

The computation domain, boundary conditions, and blocks are set in Figure 3. The geometric scale ratio of the main deck section is 1:70. The height of the grid on the first layer is 0.2 cm, which is about $4 \times 10^{-5} B$, and there are 312580 grids in total. Figure 12 indicates the local division of the grids near the main deck section of LDYB. The wind speed ratio is 1:4.16, the incoming wind speed is $U = 18$ m/s for the Volterra kernels identification, and the corresponding wind speed at the bridge is 75 m/s, which is close to the flutter critical wind speed of the LDYB section for subsequent flutter analysis. The rest of the calculation parameters are set the same as in the case of GBEB.

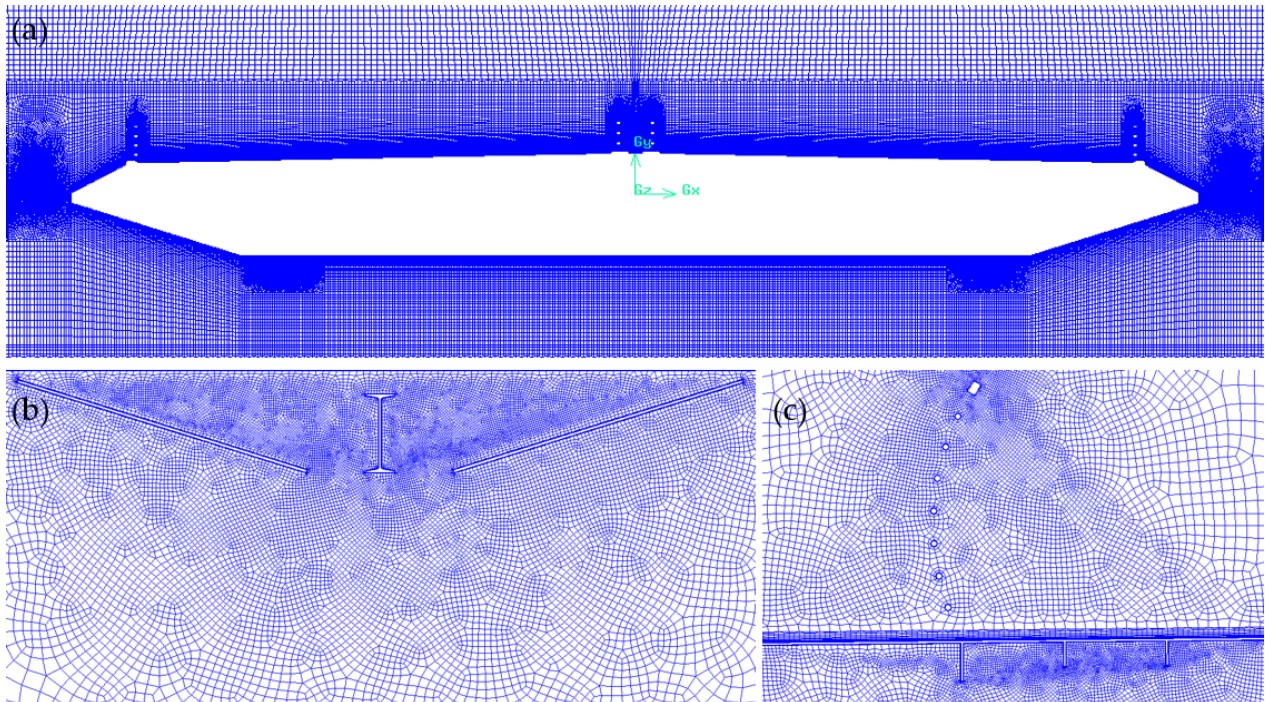

**Figure 12.** Local mesh division of the computational domain (**a**) near the main deck section; (**b**) near the maintenance vehicle tracks; and (**c**) near the sidewalk outriggers.

To study the first-order Volterra model in more nonlinear aerodynamic conditions, the wind-induced vibration responses of the main deck section of LDYB for different wind speeds near the flutter critical wind speed are calculated by ROM and FSI, respectively. The main parameters of the main deck section of LDYB are given in Table 5.

**Table 5.** Main parameters of the main deck section of LDYB.

| Parameters | Unit | Prototype Main Deck Section | Scale Ratio | Main Deck Section Model |
|---|---|---|---|---|
| Width, $B$ | m | 49.7 | 1:70 | 7.10 |
| Height, $H$ | m | 4.4 | 1:70 | 0.057 |
| Mass per unit length, $m$ | kg/m | $6.95 \times 10^4$ | $1:70^2$ | 14.184 |
| Torsion mass moment of inertia per unit length, $I$ | kg•m$^2$/m | $1.16 \times 10^7$ | $1:70^4$ | 0.483 |
| Vertical bending frequency, $f_h$ | Hz | 0.103 | 16.83:1 | 1.7334 |
| Torsional frequency, $f_\alpha$ | Hz | 0.225 | 16.83:1 | 3.7842 |
| Vertical bending–damping ratio, $\zeta_h$ | % | 0.5 | / | 0.5 |
| Torsional damping ratio, $\zeta_\alpha$ | % | 0.5 | / | 0.5 |
| Wind velocity, $U$ | m/s | / | 1:4.16 | / |

Figure 13 shows the numerical results of wind-induced vibration responses of the main deck section by ROM method for incoming wind speeds of 75 m/s and 80 m/s for the prototype main deck section, respectively. As shown in Figure 13, when the incoming wind speed is 75 m/s for the prototype main deck section, the torsional displacement and vertical displacement of the prototype girder have no obvious flutter divergence phenomenon. However, when the incoming wind speed is 80 m/s for the prototype main deck section, the torsional displacement and vertical displacement of the girder have an apparent flutter divergence phenomenon. Therefore, the calculation result of the flutter critical wind speed of the main girder section of LDYB obtained by the ROM method is about 75~80 m/s.

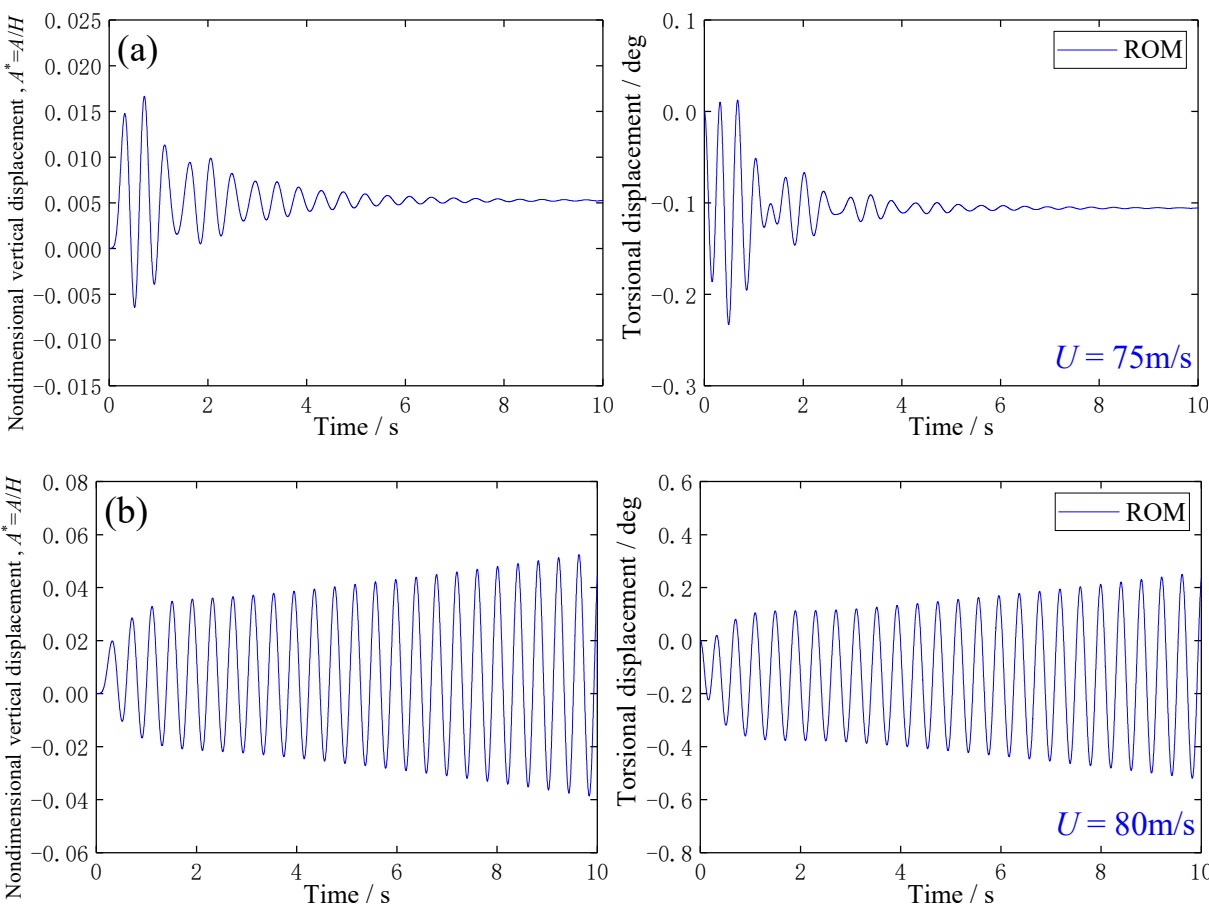

**Figure 13.** Results of flutter analysis of LDYB based on the ROM method under incoming wind speeds of (**a**) 75 m/s and (**b**) 80 m/s for the prototype main deck section, respectively.

Table 6 summarizes the flutter critical wind speed and flutter frequency results of the main girder of LDYB calculated by the aeroelastic analysis method in this study and other examples of research. It can be seen that the flutter critical wind speed of the girder of LDYB obtained by different methods is similar, while there is a slight difference between the flutter frequency. This result indicates that the aeroelastic method in this paper can simulate the flutter critical wind speed, even though the first-order Volterra model may underestimate the aerodynamic stiffness in a highly nonlinear aerodynamic system.

**Table 6.** Comparisons of flutter results of the girder of LDTB.

| Method | Flutter Critical Wind Speed (m/s) | Flutter Frequency (Hz) |
|---|---|---|
| Wind tunnel test (Wang [32]) | 76.5 | 0.198 |
| FSI (Wang [32]) | 70.0~75.0 | 0.176 |
| ROM (present) | 75.0~80.0 | 0.160 |

## 4. Discussion and Conclusions

In this paper, the main deck sections of the GBEB and LDYB are taken as the research objects, and the aerodynamic ROM is established to propose a new analysis method for the aeroelastic effect based on the Volterra series and the Newmark-$\beta$ method. The flutter critical wind speed and the flutter frequency obtained by the new aeroelastic analysis and other methods are found to be similar, thus verifying the feasibility and accuracy of the method in this paper. Although there is a certain difference between the flutter displacement response calculated by the ROM method proposed in this study and FSI with

ANSYS Fluent, the issue can be theoretically alleviated by introducing multi-order Volterra kernels, which can consider more aerodynamic nonlinear effects. Furthermore, the Volterra series-based aerodynamic ROM is suitable to be extended for the three-degrees-of-freedom flutter problem.

By comparing the flutter critical wind speed and flutter frequency obtained from the aerodynamic ROM based on the Volterra series with those from CFD numerical simulations under both forced and free vibrations, the following conclusions can be drawn:

1. The aerodynamic ROM of the main girder section is established based on the first-order Volterra series through the CFD numerical simulation. Moreover, the aerodynamic identification accuracy of the main girder is found to be in good agreement with the results of the forced vibration based on CFD numerical simulation. Additionally, the established aerodynamic ROM can consider the nonlinear aerodynamic effect caused by the certain amplitude and frequency of the streamlined main beam section.
2. An aeroelastic analysis method for the main girder section is established based on the Volterra series-based aerodynamic ROM and the Newmark-$\beta$ method, which enabled efficient FSI calculations and the determination of the flutter critical wind speed and flutter frequency. Moreover, the accuracy and feasibility of the methods in this study are verified by comparing the numerical results of different main deck sections obtained from FSI simulations based on ANSYS Fluent and the existing numerical and experimental results.

**Author Contributions:** Conceptualization, Z.W., Z.L. and F.H.; methodology, Z.W. and Z.L.; software, Z.W.; validation, Z.W. and Z.L.; formal analysis, F.H.; investigation, Z.W.; resources, Z.L.; data curation, F.H.; writing—original draft preparation, Z.W.; writing—review and editing, Z.L.; visualization, F.H.; supervision, Z.L.; project administration, Z.L.; funding acquisition, Z.L. All authors have read and agreed to the published version of the manuscript.

**Funding:** This research was funded by the National Natural Science Foundation of China, grant number 52178475 and 51778225.

**Acknowledgments:** We deeply appreciate the assistance provided by Zhijun Jiang, Zinan Lin, Han Xiao, Ruilin Zhang, Yaoheng Feng, Wei Zhou, Minghe Li, Zhenyu Gao, and Yibo Wei in the process of article writing, and the technical support by Hunan Provincial Key Lab of Wind Engineering and Bridge Engineering, Hunan University.

**Conflicts of Interest:** The authors declare no conflict of interest. The funders of the National Natural Science Foundation of China provided authors with the access to the efficient computing resources.

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
