# Peer review of "Reduced-Order Model Based on Volterra Series for Aerodynamics of the Bridge Deck Section and Flutter Critical Wind Speed Prediction"

_applsci, doi:10.3390/app13063486_

Round 1

Reviewer 1 Report

The paper is globally well organized but some aspects deserve more attention. The results show the method provides reliable predictions for the case study selected. However, in the reviewer's opinion the validation is very limited and the following main issues should be addressed before the article can be published.

1) The model is validated only in one special case. Moreover, the selected cross-section is a streamlined one, which implies a linear unsteady aerodynamics (for small-amplitude displacements, linear flutter derivatives work well; aerodynamic nonlinearities are very limited). 

2) The selected benchmark case is not a good choice to test the model in catching the nonlinear unsteady aerodynamics: in addition to the presented example, the Authors are thus invited to analyze at least another case with a more pronounced aerodynamic nonlinearity (e.g., a truss girder section / double-deck section / adding the barriers to the GBEB section / considering larger vibration amplitudes...). 

The Authors are invited to comment on the previous points, revise the Conclusions accordingly, and add at least another verification example which could better enphasize the capability of the proposed method to catch the aerodynamic nonlinearities. In the present form, it seems that the method is presented as a nonlinear reduced-order-model, which is then tested on a linear or nearly linear aerodynamic problem.

Minor issue:

Lines 145-147 appear to be out of context.

Author Response

We thank the reviewer for his/her comments, which certainly help to improve the paper. Here are detailed answers

Answer to the Reviewer #1

General comment: The paper is globally well organized but some aspects deserve more attention. The results show the method provides reliable predictions for the case study selected. However, in the reviewer's opinion the validation is very limited and the following main issues should be addressed before the article can be published.

Response: Thank the Reviewer #1 very much for the possible positive comments in our manuscript. For more validation, Lingdingyang Bridge (LDYB), a suspension bridge with main span of 1666 m under construction, is taken as another engineering background.

Comment: 1) The model is validated only in one special case. Moreover, the selected cross-section is a streamlined one, which implies a linear unsteady aerodynamics (for small-amplitude displacements, linear flutter derivatives work well; aerodynamic nonlinearities are very limited).

Response: Thank the Reviewer #1 very much for the comments and suggestions. Based on the comment and suggestion, we have test the Volterra model in the nonlinear unsteady aerodynamic (for a blunter girder section). Moreover, to more clearly illustrate this point, we have added the following sentences: “Compared with the case of the GBEB, the main deck section of LDYB includes sidewalk outriggers, crash barriers, and inspection rails, where the nonlinear aerodynamic will be more foreseeably significant.” Please see Line 303-304 in Page 12.

Comment: 2) The selected benchmark case is not a good choice to test the model in catching the nonlinear unsteady aerodynamics: in addition to the presented example, the Authors are thus invited to analyze at least another case with a more pronounced aerodynamic nonlinearity (e.g., a truss girder section / double-deck section / adding the barriers to the GBEB section / considering larger vibration amplitudes...).

Response: Thank the Reviewer #1 very much for the comment and suggestion. Based on the comment and suggestion, Lingdingyang Bridge (LDYB), a suspension bridge with main span of 1666 m under construction, is taken as another engineering background considering that the drawings of the GBEB section added the barriers are difficult to obtain. We wish that the Reviewer #1 can approve it. Moreover, the aeroelastic analysis method is applied to the prediction of the wind-induced vibration responses of the LDYB girder near the flutter critical wind speed. Please see Line 296-338 in Page 12-14 (Sect. 3.3).

The relevant content of the Abstract and Introduction has also been adjusted accordingly. Finally, the original Sect. 3.2. and 3.3. have been merged to ensure the integrity of the paper and the title of Sect. 3.2. has been changed into “Numerical results of wind-induced vibration responses”.

Comment: 3) Lines 145-147 appear to be out of context.

Response: Thank the Reviewer #1 very much for the comment and suggestion. Based on the comment and suggestion, the content has been adjusted while it was intended to introduce the fluid governing equations. We are sorry for this errors. We have added the following sentences: “For incompressible viscous fluids, the mass conservation equation and the momentum conservation equation can be expressed as follows,

where ρ denotes the fluid density; V denotes the velocity vector; p denotes the pressure; f denotes the external force on the unit volume fluid, if gravity is considered, f=ρg; μ denotes the dynamic viscosity.” Please see Line 132-137 in Page 4.

Reviewer 2 Report

The manuscript does not offer sufficiently novel research results and no new contribution. The research problem in the paper does not seem to be motivated by a clearly outlined research question and no physical insight is provided for this theoretical analysis. Therefore, it does not have a suitable reason for this study. Too many typos and grammatical errors are made as well as the qauilty of figures are very bad and poor. Thus, the results might be not useful. And my suggestion is to reject the paper.

Author Response

Answer to the Reviewer #2

We thank the reviewer for his/her comments, which certainly help to improve the paper. Here are detailed answers

General comment: The manuscript does not offer sufficiently novel research results and no new contribution. The research problem in the paper does not seem to be motivated by a clearly outlined research question and no physical insight is provided for this theoretical analysis. Therefore, it does not have a suitable reason for this study. Too many typos and grammatical errors are made as well as the quality of figures are very bad and poor. Thus, the results might be not useful. And my suggestion is to reject the paper.

Response: Thank the Reviewer #2 very much for the comments on our manuscript. Based on the comments, we have made the following endeavor:

(1) Manuscript: we have basically re-written the whole manuscript, include modeling, equation, and organization, Figures, and English. We believe that the present version is significantly improved.

 (2) Novelty: we have added the following sentence to highlight the novelty in the Introduction: “Meanwhile, there is a lack of aerodynamic self-excited force models considering vertical-torsional coupling since most current aerodynamic self-excited force models are used for single-degree-of-freedom flutter analysis. It is necessary to conduct further research on the aeroelastic analysis for the FSI calculation. More importantly, taking into account the advantage of multiple-degree-of-freedom and the convenient identification of the first-order kernel, it is of great value and significance for the development of long-span bridges to carry out the rapid analysis method research of two-degree-of-freedom flutter based on Volterra model.” Please see Line 74-79 in Page 2.

(3) To improve the English and correct the errors, we have re-written the manuscript carefully, include modeling, equation, and organization. Furthermore, we have invited a professional polishing company to polish and revise the grammar and sentences of the manuscript. We hope that the revised manuscript can meet the requirements of publication.

Reviewer 3 Report

This paper predicts the flutter critical wind speed of a bridge deck using the first-order Volterra series. The topic is interesting. The paper is well organized and written. The paper may be recommended for publication after addressing the following issues:

1. The novelty and motivation of this study are not clear. The authors should explain what is the knowledge gap filled by this study in the field of bridge aerodynamic or flutter analysis.

2. Line 145-147, these contents seems strange.

3. The quality of some figures, e.g., Figure 6, could be improved.

4. The authors utilized a first-order model. It seems the first-order Volterra model is the same as the model based on linear convolution. It is also not clear if ther first-order model is able to consider any nonlinear effect. Some discussions are suggested.

5. There are other models proposed for post-flutter analysis in recent years, e.g., the polynomial model and the describing function-based model. The authors are suggested to enrich the literature review with recent work on post-flutter analysis. Some examples include:

Postflutter analysis of bridge decks using aerodynamic-describing functions

Tuned mass damper for self-excited vibration control: Optimization involving nonlinear aeroelastic effect

Nonlinear post-flutter bifurcation of a typical twin-box bridge deck: Experiment and empirical modeling

Author Response

Answer to the Reviewer #3

We thank the reviewer for his/her comments, which certainly help to improve the paper. Here are detailed answers

General comment: This paper predicts the flutter critical wind speed of a bridge deck using the first-order Volterra series. The topic is interesting. The paper is well organized and written. The paper may be recommended for publication after addressing the following issues:

Response: Thank the Reviewer #3 very much for this positive comment on the novel idea in our manuscript.

Comment: 1. The novelty and motivation of this study are not clear. The authors should explain what is the knowledge gap filled by this study in the field of bridge aerodynamic or flutter analysis.

Response: Thank the Reviewer #3 very much for these comments. Compared with some other aerodynamic model, the Volterra model established in this paper can enable rapid analysis of flutter response, which considers the vertical-torsional coupling effect. To more clearly illustrate this, we have added the following sentences in the Introduction: “Additionally, current aerodynamic self-excited force models are mainly used for single-degree-of-freedom flutter analysis. There is a lack of models considering the vertical–torsional coupling. Therefore, further investigation into the aeroelastic analysis of the bridge deck section is necessary. Moreover, taking into account the advantage of multiple-degrees-of-freedom and the convenient identification of the first-order kernel, it is of great value and significance for the development of long-span bridges to carry out the rapid analysis method research of two-degree-of-freedom flutter based on Volterra model.” Please see Line 75-80 in Page 2.

Comment: 2. Line 145-147, these contents seems strange.

Response: Thank the Reviewer #3 very much for the comment and suggestion. Based on the comment, the content has been adjusted while it was intended to introduce the fluid governing equations. We are sorry for this errors. We have added the following sentences: “For incompressible viscous fluids, the mass conservation equation and the momentum conservation equation can be expressed as follows,

where ρ denotes the fluid density; V denotes the velocity vector; p denotes the pressure; f denotes the external force on the unit volume fluid, if gravity is considered, f=ρg; μ denotes the dynamic viscosity.” Please see Line 132-137 in Page 4.

Comment: 3. The quality of some figures, e.g., Figure 6, could be improved.

Response: Thank the Reviewer #3 very much for these comments. Based on these comments, we have improved the quality of Figure 6, which is shown in the following figure.

Figure. Schematic representation of the numerical simulation based on the Volterra model.

Comment: 4. The authors utilized a first-order model. It seems the first-order Volterra model is the same as the model based on linear convolution. It is also not clear if the first-order model is able to consider any nonlinear effect. Some discussions are suggested.

Response: Thank the Reviewer #3 very much for the comment and suggestion. Based on the comment and suggestion, we have discussed if the first-order model is able to consider the nonlinear effect. The first-order kernel is entirely linear when it is directly identified by the output response under a unit impulse, that is, h1(t)=y1(t). However, the first-order kernel is identified considering the nonlinear effect from different amplitudes, which can be seen in Equation (3). To more clearly illustrate this, the sentences: “For a first-order truncated Volterra series, Equation (1) can be rewritten as Equation (2), and the expression of the first-order kernel is shown in Equation (3)” has been replaced with “For a first-order truncated Volterra series, Equation (1) can be rewritten as Equation (2), and the first-order kernel can be directly expressed as the output response under the action of a unit impulse equation, that is, h1(t)=y1(t). Such a first-order kernel is entirely linear. Therefore, Equation (3) is utilized to identify the first-order Volterra kernel. It can be seen that the first-order Volterra kernel obtained by Equation (3) considers the nonlinear effect of different amplitudes, which is different from the general linear kernel.” Please see Line 99-104 in Page 3.

Comment: 5. There are other models proposed for post-flutter analysis in recent years, e.g., the polynomial model and the describing function-based model. The authors are suggested to enrich the literature review with recent work on post-flutter analysis. Some examples include:

Postflutter analysis of bridge decks using aerodynamic-describing functions

Tuned mass damper for self-excited vibration control: Optimization involving nonlinear aeroelastic effect

Nonlinear post-flutter bifurcation of a typical twin-box bridge deck: Experiment and empirical modeling

Response: Thank the Reviewer #3 very much for the comment and suggestion. Based on the comment and suggestion, the literature review has been enriched. Considering the rigor of the logic of the Introduction, other models proposed for post-flutter analysis in recent years are discussed as comparisons of the Volterra model in Sect. 2.4. We wish that the Reviewer #3 can approve it. To more clearly illustrate this, we have added the following sentences: “In this case, the Volterra series-based ROMs are models of four first-order Volterra kernels to be identified. Accordingly, due to the advantage of fewer parameters to be solved, the computational cost to establish the ROMs can be greatly reduced compared with flutter derivative theory and some other flutter analysis methods, e.g., aerodynamic describing-function-based models (Zhang et al. [27]). Moreover, the vertical–torsional-coupling effect has been considered, significantly improved over single-degree-of-freedom (SDOF) torsional flutter simplified models (e.g., Gao et al. [28]).” Please see Line 182-187 in Page 6.

Round 2

Reviewer 1 Report

The Authors have revised the paper according with the reviewer's remarks and suggestions. The present revised version results improved, the main lacs of the original version being overcome. The paper is now globally more sound, even if it it could still be improved. Thus, the reviewer's suggestion is to accept the paper after minor revision (without need of re-check by the reviewer), after re-reading for minor text editing and possible refinements.

Author Response

Answer to the Reviewer #1

We thank the reviewer for his/her comments, which certainly help to improve the paper. Here are detailed answers

General comment: The Authors have revised the paper according with the reviewer's remarks and suggestions. The present revised version results have been improved, and the main lacs of the original version have been overcome. The paper is now globally more sound, even if it could still be improved. Thus, the reviewer's suggestion is to accept the paper after minor revision (without need of re-check by the reviewer), after re-reading for minor text editing and possible refinements.

Response: Thank the Reviewer #1 very much for the possible positive comments on our manuscript. Based on the comments and suggestions, we have text edited the Discussion and conclusions (Sect. 4.) and modified the original grammatical error. 

Reviewer 3 Report

The reviewer thanks the authors for addressing the comments. The paper is recommended to publish in the present form.

Author Response

Answer to the Reviewer #3

We thank the reviewer for his/her comments, which certainly help to improve the paper. Here are detailed answers

General comment: The reviewer thanks the authors for addressing the comments. The paper is recommended to publish in the present form.

Response: Thank the Reviewer #3 very much for this positive comment in our manuscript.